# Fast Coordinated Predictive Control for Renewable Energy Integrated Cascade Hydropower System Based on Quantum Neural Network

**Xi Ye** [1], **Zhen Chen** [2], **Tong Zhu** [1], **Wei Wei** [2] and **Haojin Peng** [3],*

1   State Grid Sichuan Electric Power Company, Chengdu 610041, China; yex3825@sc.sgcc.com.cn (X.Y.); zhut0158@sc.sgcc.com.cn (T.Z.)
2   Electric Power Research Institute, Chengdu 610041, China; chenz5518@sc.sgcc.com.cn (Z.C.); weiw1216@sc.sgcc.com.cn (W.W.)
3   College of Electrical Engineering, Sichuan University, Chengdu 610065, China
*   Correspondence: penghaojin@stu.scu.edu.cn; Tel.: +86-19961651881

**Abstract:** The increasing penetration of renewable energy poses intractable uncertainties in cascade hydropower systems, such that excessively conservative operations and unnecessary curtailment of clean energies can be incurred. To address these challenges, a quantum neural network (QNN)-based coordinated predictive control approach is proposed. It manipulates coordinated dispatch of multiple clean energy sources, including hydro, wind, and solar power, leverages QNN to conquer intricate multi-uncertainty and learn intraday predictive control patterns, by taking renewable power, load, demand response (DR), and optimal unit commitment as observations. This enables us to exploit the stability and exponential memory capacity of QNN to extrapolate diversified dispatch policies in a reliable manner, which can be hard to reach for traditional learning algorithms. A closed-loop warm start framework is finally presented to enhance the dispatch quality, where the decisions by QNN are fed to initialize the optimizer, and the optimizer returns optimal solutions to quickly evolve the QNN. A real-world case in the ZD sub-grid of the Sichuan power grid in China demonstrates that the proposed method hits a favorable balance between operational cost, accuracy, and efficiency. It realizes second-level elapsed time for intraday predictive control.

**Keywords:** intraday predictive control; renewable energy integrated cascade hydropower system; quantum neural network; closed-loop warm start

## 1. Introduction

Stochastic optimization (SO) [1] and robust optimization (RO) [2] methods are commonly employed for modeling and addressing the coordinated scheduling of complementary power generation systems with multi-dimensional uncertainties. These uncertainties include various renewable energy sources, such as wind, solar, and water resources [3,4]. However, SO and RO face challenges in accurately describing the probability distribution regulation of uncertainties in hybrid power systems, thereby leading to reduced reliability [5]. Additionally, SO relies on a large number of discrete scenarios, resulting in computationally intensive processes and lengthy model solving time. In contrast, RO does not require predefined probability distributions for random variables [6–8]. Nevertheless, because RO seeks optimal solutions based on worst-case scenarios [9], it tends to yield overly conservative optimization results, limiting resource utilization in multi-energy complementary scheduling. Furthermore, neither of these algorithms can achieve fast online solutions for complex systems. The modeling of multi-energy complementary systems introduces a multitude of variables and constraints, resulting in a high degree of model complexity. The high model complexity makes it challenging to achieve fast real-time solutions for SO and RO through traditional solving methods, such as C&CG [10], Benders [11], and heuristic algorithms [12].

The shortcomings of classical methods, such as SO and RO, in terms of solution efficiency pose challenges for real-time applications in predictive control. Even with methods based on neural networks, two main drawbacks persist for large-scale power systems. Firstly, there is a risk of losing crucial variable characteristic information. Secondly, a substantial amount of matrix operations is required during the training phase to update network parameters through the backpropagation algorithm. As deep neural networks have more parameters, these computations become more complex and time-consuming.

With the rapid advancement of artificial intelligence technology, it has found widespread applications in renewable energy forecasting, load forecasting, and intelligent control of operational modes. Among these approaches, neural network methods demonstrate strong capabilities in handling high-dimensional nonlinear data [13–16]. Trained neural network models exhibit robust online applicability. For instance, the backpropagation (BP) neural network has been applied to short-term load forecasting [17], short-term economic dispatching [18], and power flow forecasting [19]. Ref. [20] characterized the complex mapping relationship between real-time operation characteristics of the power grid and optimal nonconvex economic dispatching solutions by constructing an integrated deep neural network. Reference [21] proposed the use of neural networks to solve an economic dispatching method considering transmission line capacity constraints. Ref. [22] employed a deep neural network (DNN) method to address the DC optimal power flow problem with security constraints (SC-DCOPF). It is worth noting that the above-mentioned research has been conducted within the limited scale and system complexity. In large-scale complex systems, when neural networks are converted into differentiable forms, the existence of a significant number of state variables may result in information loss. Furthermore, the model training process becomes computationally intensive and time-consuming.

Quantum computing [23–25], which incorporates the superposition principle of quantum states, possesses parallel computing capabilities. In comparison to traditional neural networks, quantum neural networks [26–28] offer a higher storage capacity, effectively mitigating the vanishing gradient problem. This makes them particularly well-suited for high-demand computations involving large-scale datasets. Ref. [29] proposed a radial basis function (RBF) neural network for power transformer fault diagnosis based on a hybrid adaptive training method. They employed a quantum particle swarm optimization (QPSO)-based RBF neural network model to automatically optimize the network structure and obtain model parameters, thereby improving the classification accuracy. Ref. [30] developed a quantum transient stability assessment (QTSA) method based on quantum generator learning, enabling efficient prediction of large power grid transient stability through data-driven approaches. This method simplifies the handling of transient stability assessment (TSA) in Hilbert space. Ref. [31] employed QNN training to implement an indirect adaptive fuzzy wavelet neural controller as a power system stabilizer to suppress inter-area oscillation modes in multi-generator power systems, effectively enhancing the computational efficiency of neural networks through quantum computing. Ref. [32] proposed a hybrid quantum LSTM model for one-hour-ahead solar irradiance prediction, which combines VQC and LSTM to obtain richer time-dependent information in meteorological and solar radiation data. The powerful computational capabilities of quantum neural networks overcome the challenges of fast predictive control, while also demonstrating the ability for quick and efficient model training. Additionally, quantum neural networks exhibit higher storage capacity and superior capabilities in handling large-scale state spaces, making them less prone to information loss. This ensures the accurate mapping ability of the model.

Based on the aforementioned research, this paper proposes a method for achieving real-time predictive control of complex systems using a QNN in the context of a multi-energy system. Firstly, we construct an operational scheduling model encompassing various clean energy sources such as water, wind, and solar energy. We also consider demand response [33,34] to enhance the model's peak-shaving capabilities and the coordinated integration of renewable energy. Subsequently, leveraging historical measured data, we train a QNN prediction model. Finally, we establish a joint optimization framework

integrating the quantum neural network and the multi-energy system scheduling model. This framework involves the transfer of QNN prediction data to the scheduling model, facilitating hot start, and the propagation of solution results back to the QNN to promote evolution. This constitutes a closed-loop learning optimization framework, enabling rapid predictive control within a daily timeframe. The proposed method has been validated in the context of Sichuan Zhengdou.

## 2. Intraday Predictive Control Fusion Optimization Method Based on QNNs

### 2.1. Optimal Dispatching Model Considering Multi-Energy Complementary Coordination

The dispatching model proposed in this paper, which takes into account the coordination of multiple energy sources, should consider the coordinated operation cost of hydropower, wind energy, and solar energy, as well as the operating power purchase cost. The model objective is as follows:

$$min C = \sum_{t=1}^{T} \left( C_t^{\text{buy}} + C_t^H + C_t^{IIG} + C_t^{DR} \right) \tag{1}$$

where $C_t^{\text{buy}}$, $C_t^H$, $C_t^{IIG}$, and $C_t^{DR}$ are, respectively, the power purchase cost, the operation cost of the cascade hydropower system, wind/photovoltaics power station, and demand response in the period $t$. In this paper, $C_t^{DR}$ takes into account the load shedding costs, which is determined based on the supply–demand relationship. The cost is calculated by multiplying the reduced quantity by the unit reduction cost.

The constraints of the model include the formation of operational constraints based on the predicted information of cascade hydropower systems and wind/photovoltaics systems, mainly including power balance, constraints on water balance, unit operation, and power grid.

Formula (2) represents the power balance constraint of the power grid:

$$\sum_{o=1}^{\Omega} P_{o,t} + \sum_{h=1}^{N^H} P_{h,t}^H + \sum_{r=1}^{N^{IIG}} P_{r,t}^{IIG} = \sum_{l=1}^{N^L} P_{l,t}^L - \sum_{c=1}^{N^C} P_{c,t}^{DR} \tag{2}$$

where $\Omega$ represents the power purchase status parameters during time $t$. In the period $t$, $P_{o,t}$, $P_{h,t}^H$, $P_{r,t}^{IIG}$, $P_{l,t}^L$, and $P_{c,t}^{DR}$ represent the purchased power, the power generation of the $h$-th cascade hydropower unit, the power generation of the $r$-th renewable energy unit, the active power of the $l$-th load, and the shedding amount of the $c$-th load, respectively.

The model's standard constraints consist of cascade hydropower system constraints, power grid operational constraints, and renewable energy constraints. Within the cascade hydropower constraints, there are reservoir capacity constraints, flow constraints, and output constraints. The renewable energy constraints encompass output constraints for wind/photovoltaics units. Specific constraints are detailed in Appendix A.

The power flow constraint of the power grid involves power flow calculation. Power flow calculation in optimization can be divided into AC power flow and DC power flow. The AC power flow method is complex in calculation and difficult to quickly solve in optimization. Since the resistance of tie lines in transmission lines is often far less than the line reactance value, it is easier to use the DC power flow method to solve the power flow of critical inter-corridors, and its constraints are as follows:

$$P_{\text{tline}} = B_{\text{diag}} L B^{-1} \left( P_t + P_t^{IIG} - P_t^L - P_t^{DR} \right) \tag{3}$$

$$-\overline{P}_{\text{line}} \leq P_{\text{tline}} \leq \overline{P}_{\text{line}} \tag{4}$$

$$B_{\text{diag}} = diag \left( \frac{1}{x_1}, \frac{1}{x_2}, \cdots, \frac{1}{x_N} \right) \tag{5}$$

where $P_{\text{tline}}$ is the DC power flow power of each branch; $B$, $L$ are the admittance coefficient matrix and node connection matrix of the branch in the network respectively; $P_t$, $P_t^H$, $P_t^{IIG}$,

$P_t^L$, and $P_t^{DR}$ are the vector forms of the purchased power, the output power of the cascade hydropower system, the output power of wind/photovoltaics power station, the load active power demand, and the load shedding amount in period $t$, respectively; $\overline{P}_{\text{line}}$ is the maximum transferable branch power; $x_l$ is the reactance of the branch; $N$ is the branch number in the network.

In demand response, the load reduction capacity is subject to capacity constraints, as illustrated below.

$$P_{c,min}^{DR} \leq P_{c,t}^{DR} \leq P_{c,max}^{DR} \tag{6}$$

where $P_{c,min}^{DR}$ and $P_{c,max}^{DR}$ represent the lower and upper bounds of the load reduction capacity, respectively.

*2.2. Intraday Operational Information Prediction Model Based on QNNs*

In the realm of power systems, devising scheduling strategies rooted in optimization theory encounters a challenge due to the "dimensional explosion" issue arising from the high penetration of renewable energy sources. Nevertheless, quantum computing provides a promising solution by efficiently handling large-scale datasets. Additionally, quantum neural networks exhibit superior computational capabilities for high-dimensional features in comparison to traditional neural networks.

In this paper, we present a quantum neural network prediction model to tackle these challenges. Our approach involves constructing a mapping model that takes real-time measurements of wind, photovoltaics, and load as input data. The model's outputs include unit output and unit commitment. Utilizing this model, we facilitate the initiation of real-time "warm start" optimization for intraday scheduling decision variables. With the application of the quantum neural network prediction model, our goal is to overcome the constraints imposed by the dimension explosion issue and facilitate the development of efficient and effective intraday control strategies.

2.2.1. Quantum Theory

In quantum theory, the state of a quantum bit can be expressed as follows:

$$| \Psi \rangle = a_0 \, | \, 0 \rangle + a_1 \, | \, 1 \rangle \tag{7}$$

where $a_0$ and $a_1$ are any complex numbers which meet the normalization requirements $| \, a_0 \, |^2 + | \, a_1 \, |^2 = 1$, and $| \, a_0 \, |^2$ and $| \, a_1 \, |^2$, respectively, represent the probability of the quantum bit collapsing to $| \, 1 \rangle$ and $| \, 0 \rangle$. The probability of the quantum state collapsing to $| \, 1 \rangle$ is taken as the imaginary part, and the probability of the quantum bit collapsing to $| \, 0 \rangle$ is taken as the real part, which is expressed in complex form, as in Formula (8):

$$f(\theta) = \cos(\theta) + j\sin(\theta) \tag{8}$$

where the probability amplitude of $| \, 0 \rangle$ corresponds to the square of the real part. The $| \, 1 \rangle$ probability amplitude corresponds to the square of the imaginary part; $j$ is an imaginary unit; $\theta$ is the phase angle, and different phase angles correspond to different quantum states.

Quantum gates serve as the fundamental building blocks for implementing quantum computing. They embody the defining characteristics of quantum computing itself. Based on Formula (8), the two primary types of quantum gates that form the fundamental universal quantum gate group can be expressed as follows:

One-phase sliding door:

$$f(\theta_1 + \theta_2) = f(\theta_1)f(\theta_2) = e^{j(\theta_1 + \theta_2)} \tag{9}$$

Two-bit controlled NOT gate:

$$f[(\pi/2)k - \theta] = \begin{cases} \sin\theta + j\cos\theta, & k = 1 \\ \cos\theta - j\sin\theta, & k = 0 \end{cases} \tag{10}$$

where $k$ is the control variable, and when $k = 1$, the quantum state will be inverted; do not reverse when $k = 0$. Different $\theta$ correspond to different quantum states, and two types of quantum gates change the $\theta$ value, which realizes the evolution and transformation of quantum states.

In quantum computation, a quantum state is finally transformed into classical information (a certain value) in the form of probability amplitude through quantum measurement, namely quantum collapse. $\mid \psi \rangle$ takes probability $\mid a_0 \mid^2$ as the quantum measurement of quantum zero state, and $\mid \psi \rangle$ takes probability $\mid a_1 \mid^2$ as the quantum measurement of quantum one state.

### 2.2.2. Quantum Neural Network Architecture

The structure of quantum neural networks can be generalized as follows:

$$\pi(\boldsymbol{x},\theta) = \frac{1}{2}(1 + \langle \psi(\boldsymbol{x}) \mid U(\theta)^\dagger Z_{n-1} U(\theta) \mid \psi(\boldsymbol{x}) \rangle) + \boldsymbol{b} \tag{11}$$

where $\pi(\boldsymbol{x},\theta)$ is the output of the network, and $\psi(\boldsymbol{x})$ is the input quantum state. After the input $\boldsymbol{x}$ is encoded as a quantum state, it is applied to a series of quantum gates $U(\theta)$. $Z$ represents the $Z$ gate in quantum computing, and $\boldsymbol{b}$ is the bias term.

In this paper, we propose a quantum neural network (QNN) model that utilizes a multi-layer excitation function. The QNN architecture consists of an input layer, hidden layers, and an output layer. The input and output layers follow the same structure as a classical BP neural network. However, the hidden layers of the QNN incorporate the concept of quantum state superposition and employ multiple quantum-level transformation functions. Each quantum neuron in the hidden layer is composed of a series of excitation functions, generated by the superposition of multi-level sigmoid functions and a layer of linear functions. This design allows for the representation and processing of information at various quantum levels.

The output function of hidden layer nodes in the network structure is as follows:

$$\boldsymbol{b}_{\mathrm{r}} = \frac{1}{n_{\mathrm{s}}} \sum_{r=1}^{n_{\mathrm{s}}} f\left[\alpha_{\mathrm{n}}\left(V^{\mathrm{T}}\boldsymbol{x} - \theta^{\mathrm{r}}\right)\right] \tag{12}$$

where $\theta^{\mathrm{r}}$ is the quantum spacing; $n_{\mathrm{s}}$ is the number of quantum intervals, and the selection of its size is related to the predicted number of classifications; $\alpha_{\mathrm{n}}$ is the steepness factor; $V^{\mathrm{T}}$ is the transposition of the weights of the input layer of the neural network; $\boldsymbol{x}$ represents the input of the neural network. In this paper, $\boldsymbol{x}$ is described as follows:

$$\boldsymbol{x} = \left[\boldsymbol{x}_1;\ldots;\boldsymbol{x}_j;\ldots;\boldsymbol{x}_n\right], \forall \boldsymbol{x}_j \in \mathbb{R}^{1 \times p}, \boldsymbol{x} \subseteq \mathbb{E} \tag{13}$$

$$\boldsymbol{x}_j = \left[P_{vre}^{IIG}, P_d^L\right], \forall vre \in \mathbb{PV} \cup \mathbb{W}, d \in \mathbb{D} \tag{14}$$

where $\mathbb{E}$ represents the input sample set; $\mathbb{PV}$, $\mathbb{W}$, and $\mathbb{D}$ represent the PV unit set, wind turbine unit set, and load set, respectively. $\boldsymbol{x}_j$ contains the operation data with p-dimension characteristics; characteristics $P_{vre}^{IIG}$ and $P_d^L$ represent the intraday measured data of renewable energy and load, respectively.

The sample output $y(\boldsymbol{x})$ is the startup/shutdown status and active output level of the synchronous generators.

The gradient descent method is still used to train the quantum neural network model with multi-layer excitation functions. The differential of the loss function to parameter a can be expressed as follows (15):

$$\partial_{\theta_i} l(\boldsymbol{x},\theta) = (\pi(\boldsymbol{x},\theta) - y(\boldsymbol{x}))\partial_{\theta_i} \pi(\boldsymbol{x},\theta) \tag{15}$$

Expand the last term and take the derivative to remove the constant term:

$$\partial_{\theta_i}\pi(x,\theta) = \frac{1}{2}\partial_{\theta_i}\langle\psi(x) \mid U(\theta)^{\dagger}Z_{n-1}U(\theta) \mid \psi(x)\rangle \tag{16}$$

Further expansion and Hermitian conjugation can transform the formula into the following Formula (17):

$$\mathrm{Re}\left(\langle\psi(x) \mid (\partial_{\theta_i}U(\theta))^{\dagger}Z_{n-1}U(\theta) \mid \psi(x)\rangle\right) \tag{17}$$

where $U(\theta)$ is composed of multiple gates, and each gate is controlled by different parameters. $U(\theta)$ can be constructed as $G$ gate for derivation, and the expression of $G$ gate is as follows:

$$G(\theta) = \begin{bmatrix} \cos\theta & \sin\theta \\ -\sin\theta & \cos\theta \end{bmatrix} \tag{18}$$

In each training cycle, the training algorithm not only updates the connection weight between neurons in different layers but also updates the quantum spacing of neurons in the hidden layer. The former is the same as the conventional BP network updating algorithm, and the latter's quantum interval updating algorithm for hidden layer neurons is as follows.

By minimizing the total class variance, the update equation of the quantum interval $\Delta\theta_i^r$ of the hidden layer is obtained. The $r$ quantum intervals of the $I$ th neuron in the hidden layer are as follows:

$$\Delta\theta_i^r = \eta\frac{\lambda_k}{n_s}\sum_{m=1}^{n_s}\sum_{x_k\in C_m}\left(\tilde{\tilde{\xi}}_{im} - \tilde{\tilde{\xi}}_{ik}\right)\left(\tilde{v}_{im}^r - \tilde{v}_{ik}^r\right) \tag{19}$$

$$\tilde{\tilde{\xi}}_{im} = \frac{1}{\mid C_m\mid}\sum_{x_k\in C_m}\tilde{\tilde{\xi}}_{ik} \tag{20}$$

$$\tilde{v}_{im}^r = \frac{1}{\mid C_m\mid}\sum_{x_k\in C_m}\tilde{v}_{ik}^r \tag{21}$$

$$\tilde{v}_{ik}^r = \tilde{\tilde{\xi}}_{ik}\cdot\left(1 - \tilde{\tilde{\xi}}_{ik}\right) \tag{22}$$

where $\eta$ is the network learning rate; $\lambda_k$ is the steepness factor; $n_s$ is the number of neuron outputs; $\tilde{\tilde{\xi}}_{ik}$ is the output of the $i$ th neuron in the hidden layer when the input vector is $x_k$; $C_m$ is the pattern class vector. The structure of the QNN is depicted in Figure 1.

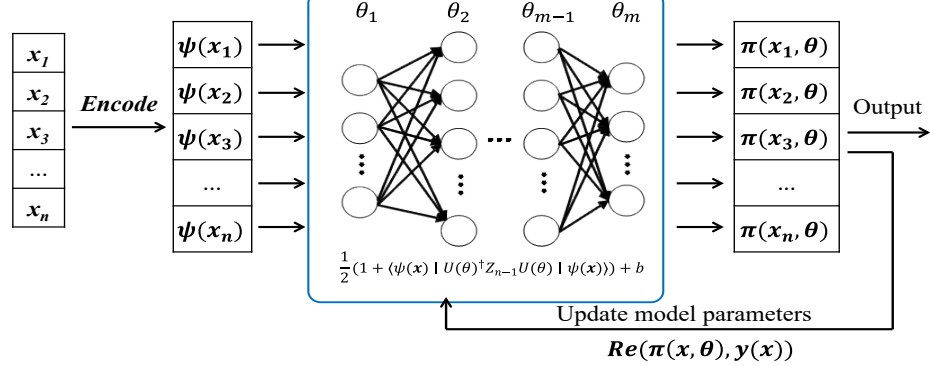

**Figure 1.** Multi-hidden layer quantum neural network structure.

### 2.3. Closed-Loop Warm Start Joint Optimization Framework Based on Quantum Neural Networks

Taking into account the joint optimization of the multi-energy system of the quantum neural network, first of all, the QNN model needs to be trained through a large number of training samples. The input characteristics of the training samples are the historical wind/photovoltaics active power output measurement data $P^{IIG}$ and the load active power measurement data $P^L$. The training label is synchronous unit startup/shutdown status $u^G$, and active output $P^G$ and $P^{DR}$.

In the prediction stage, the intraday measured data of renewable energy $P_t^{IIG0}$ and load $P_t^L$ are input into the QNN to realize the rapid prediction of synchronous generators' output and DR information $u_{t0}^G$, $P_{t0}^G$, and $P_{t0}^{DR}$. It is worth noting that the classification and regression tasks need to be realized by two QNNs to predict the $u_{t0}^G$ and $P_{t0}^G$, $P_{t0}^{DR}$.

In the online phase, we have established a closed-loop warm start framework that integrates QNN with the multi-energy system optimization model. The predictions of synchronous generators' output during the forecasting phase serve as supplementary decision-making information for intraday optimization scheduling. They play a crucial role in initializing the model and expediting its convergence towards the optimal solution. The resulting optimal solution is then fed back into the QNN model, facilitating further learning and evolution, ultimately enhancing the accuracy of the forecasting results. Using this approach, we rapidly optimize and derive intraday control strategies.

The mathematical representation o'' the scheduling initial solution mapping model constructed based on QNN is as follows:

$$\begin{cases} s = f(a) \\ IniS = s \end{cases} \tag{23}$$

$$a = \left[ P_t^{IIG0}, P_t^L \right] \tag{24}$$

$$s = \left[ u_{t0}^G, P_{t0}^G, P_{t0}^{DR} \right] \tag{25}$$

where the vector $a$ encompasses the measured data of renewable energy and load, namely $P_{r,t}^{IIG0}$ and $P_{l,t}^L$, respectively. The function $f(\cdot)$ represents the mapping model based on QNN (quantum neural network). The vector $s$ includes $u_{t0}^G$, $P_{t0}^G$, and $P_{t0}^{DR}$, which are predicted through the QNN model. It is worth noting that $G$ encompasses the output data of both hydro and conventional synchronous generators. The variable $IniS$ represents the corresponding initial solution for optimization. It is important to note that in this model, the status of units being on or off is reflected by whether the output of the synchronous generator is zero. Specifically, a value of 0 indicates that the generator unit is turned off, while a value of 1 indicates that it is turned on. The closed-loop warm start process for QNNs and the optimized model is as Figure 2:

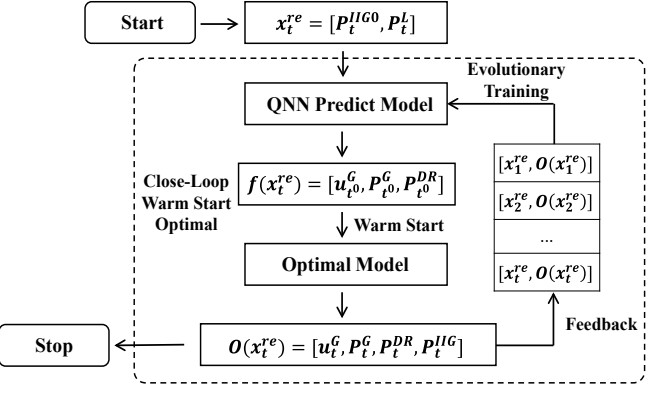

**Figure 2.** Closed-loop warm start process for QNNs and optimized model.

The QNN model utilized for predicting synchronous generator information is obtained through offline training with a substantial amount of data. This approach enables the prompt derivation of decision results for synchronous generator output and status from the measured renewable energy data. Consequently, it facilitates the acceleration of coordinated scheduling across multiple energy sources, mitigates the adverse effects of renewable energy prediction errors on scheduling outcomes, and enables rapid decision making regarding intraday power generation strategies for multi-energy complementary systems. Currently, there are various tools available for constructing QNN models, including Qiskit (based on Python), Pennylane, Tensorflow, and more. In this particular example, the QNN model was built using the Qiskit toolkit together with the Python language, with three quantum hidden layers. The overall framework for joint optimization is depicted in Figure 3.

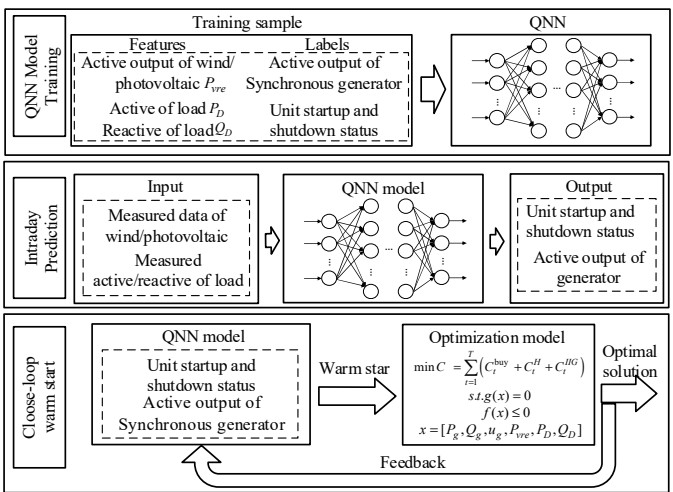

**Figure 3.** Closed-loop warm start joint optimization framework.

The multi-energy complementary coordinated dispatching model based on QNN proposed in this paper is applied in the test of Sichuan Power Grid ZD to verify the feasibility. The equivalent system of ZD is shown in Figure 4. According to the topology structure and basic data, the region is simplified into a system of 29 nodes, including power plants such as Batang Power Plant, Suwalong, Kajiwa, Yangfanggou, and Kara. The benchmark operation mode refers to the Panxi renewable energy power generation operation pattern. Under this operation pattern, Gannan is grid-connected to 1700 MW, ZD collection station is grid-connected to 600 MW, and Jinshang DC transmission power is 5 million kW.

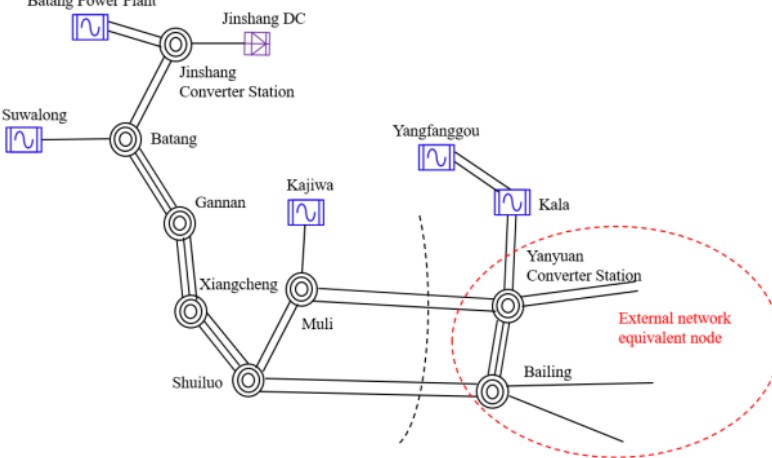

**Figure 4.** ZD near-area equivalent system.

## 3. Results

### 3.1. Results of Classification and Regression

To validate that QNNs are better suited for complex power grid scheduling problems involving binary variables like the startup and shutdown of generating units, a comparative experiment was designed between QNNs and backpropagation (BP) neural networks. In the comparison between the QNN model and the BP model for fitting the active output of synchronous units under different operating environments, Figure 5a clearly reveals a significant deviation between the generator output curve obtained from the BP network model fitting and the actual curve. However, in Figure 5b, the generator output curve resulting from fitting the QNN model exhibits a high degree of alignment with the actual curve, indicating minimal error and superior fitting accuracy.

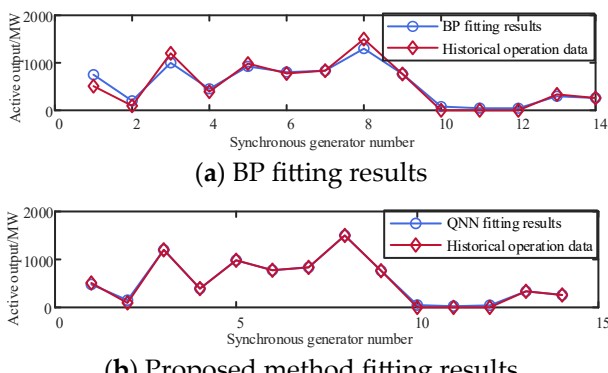

(**a**) BP fitting results

(**b**) Proposed method fitting results

**Figure 5.** Fitting effect of two neural network models.

As shown in Table 1, the classification performance of the QNN-based warm start scheduling model reached 98.91%. This represents a significant improvement of 4.57% in accuracy when compared to the traditional BP algorithm. It is noteworthy that the scheduling model based on the closed-loop warm start framework achieved an impressive classification accuracy of 99.68%. This underscores that with an ample volume of training data, this model can make precise determinations regarding unit start–stop states.

**Table 1.** BP and QNN classification accuracy.

| Model | BP | QNN Warm Start + Optimizer Decision | QNN Warm Start + Optimizer Decision + QNN Closed-Loop Evolution |
|---|---|---|---|
| accuracy | 94.34% | 98.91% | 99.68% |

It can be observed that the QNN model proposed in this paper exhibits superior performance in both classification and regression tasks compared to the traditional BP neural network.

### 3.2. Model Solving Time

Table 2 compares the solution times of the day-ahead scheduling model, the traditional RO model, the QNN-based warm start optimization model, and the QNN-based closed-loop warm start optimization model. It can be observed that the QNN-based closed-loop warm start optimization model significantly improves efficiency compared to the day-ahead scheduling model and the RO model, with a boost of 85.31% and 76.34%, respectively. Moreover, the closed-loop warm start optimization model, based on QNN, further enhances the predictive capabilities of QNN, reduces the solution space for finding the optimal solution, and results in an additional 35.9% increase in solving speed, enabling rapid online decision making.

**Table 2.** Comparison of solving times among different models.

| Model | Day-Ahead Scheduling Model | Traditional RO Model | QNN Warm Start + Optimizer Decision | QNN Warm Start + Optimizer Decision + QNN Closed-Loop Evolution |
|---|---|---|---|---|
| Solution time (s) | 358.0938 | 239.357 | 88.3415 | 52.62 |

### 3.3. Analysis of Dispatching Output Deviation of Renewable Energy Generating Units

As shown in Figure 6, a comparison is made between the measured data and the optimized output of renewable energy obtained from two different models: the day-ahead scheduling model and the QNN-based multi-energy coordinated scheduling model proposed in this paper. Specifically, the wind generators' output capacity in the ZD area is used as an example to illustrate the differences. Figure 6a demonstrates the results of the day-ahead dispatching model, which optimizes the power generation plan based on the forecasted renewable energy curve for the next day. As a result, there is a significant deviation between the predicted renewable energy output level and the actual measured data. This discrepancy arises from the substantial forecast error associated with long-term predictions.

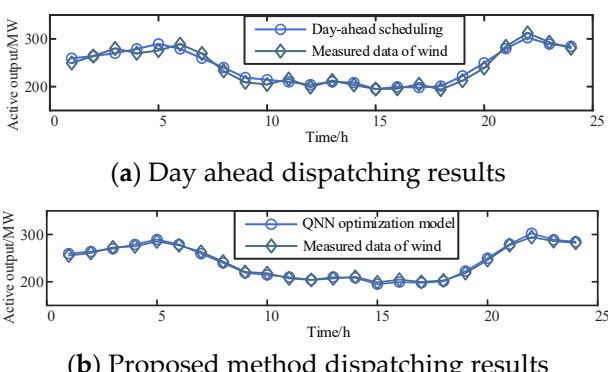

(**a**) Day ahead dispatching results

(**b**) Proposed method dispatching results

**Figure 6.** Discrepancies between scheduling results and predictions.

On the other hand, Figure 6b showcases the approach proposed in this paper, which relies on real-time measurements of the output level of renewable energy units. This method operates on a shorter time scale, leading to smaller prediction errors and scheduling deviations. Consequently, the model based on measured data yields more accurate and reliable decisions for the current moment.

These findings highlight the superiority of the QNN-based multi-energy coordinated scheduling model in minimizing prediction errors and improving the accuracy of renewable energy output optimization. By utilizing real-time measurements, the proposed model enhances the reliability and effectiveness of decision-making processes in the field of power systems.

### 3.4. Validation of the Rationality of Scheduling Results in Multi-Energy Systems

The coordinated dispatching results of the hydropower and renewable energy system are depicted in Figure 7, demonstrating the capability of the proposed model to achieve multi-energy coordination and complementarity in the ZD of the Sichuan power grid. It is noteworthy that during the 10:00–16:00 period, when the available active power of renewable energy units is relatively high, synchronous generating units will decrease their output and may even shut down certain units. Simultaneously, flexible loads participate in coordinated scheduling, facilitating the integration of renewable energy. The findings serve as evidence of the rationality and feasibility of the model's dispatching approach

within the multi-energy complementary system. Partial dispatching results are presented in Table 3, providing further insights into the outcomes.

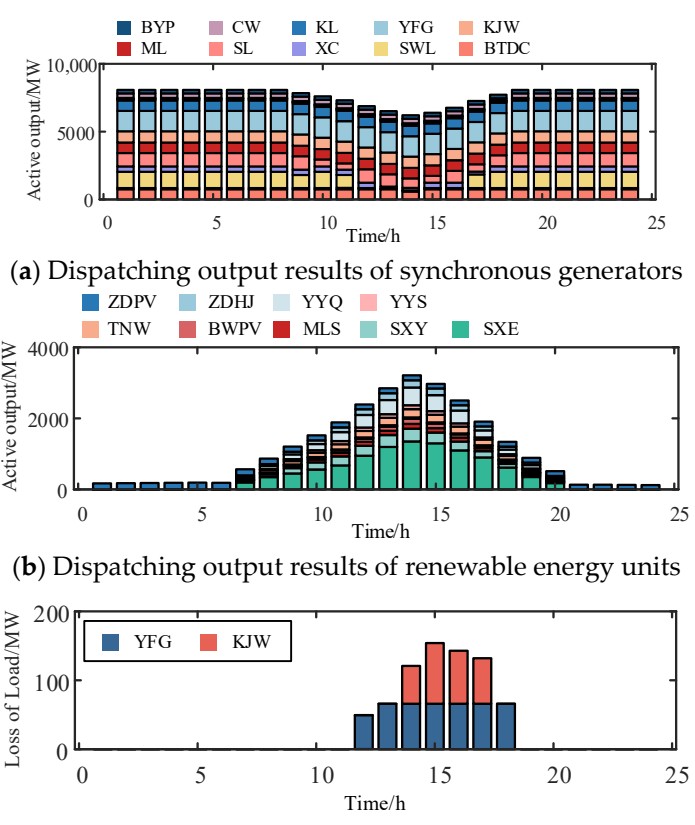

(**a**) Dispatching output results of synchronous generators

(**b**) Dispatching output results of renewable energy units

(**c**) Load shedding curve

**Figure 7.** Model optimization scheduling results.

**Table 3.** Partial dispatching results.

| Node ＼ Time | 9:00 | 12:00 | 13:00 |
|---|---|---|---|
| 9 | 834.4 | 834.4 | 834.4 |
| 10 | 1500 | 1500 | 1113.525 |
| 11 | 765 | 765 | 765 |
| 17 | 100 | 40 | 0 |
| 19 | 40 | 15 | 0 |

Figure 8 showcases a heatmap that depicts the load factor of synchronous generator units. In response to high new energy generation, a strategy has been developed to prioritize the shutdown of smaller hydroelectric units (#17, #19, and #20) based on measured data from new energy sources. This deliberate action aims to create transmission capacity and facilitate the export of renewable energy. The effectiveness of this proposed strategy in managing the integration of new energy sources into the power system is clearly demonstrated.

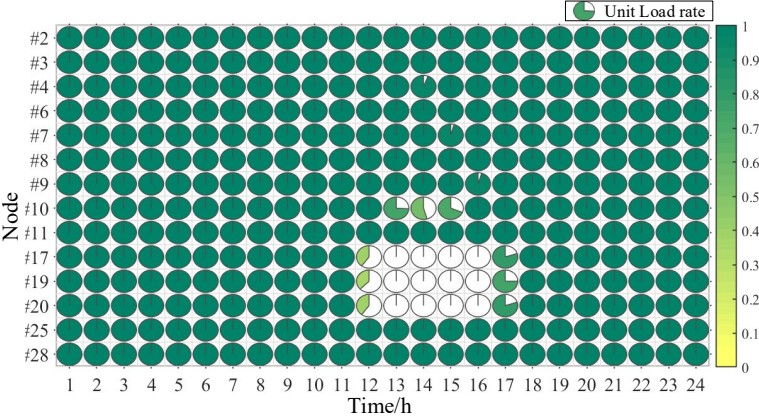

**Figure 8.** Unit load rate heat map.

*3.5. Cost Analysis*

To validate the economic feasibility of the proposed method, we conducted a comparative analysis by examining the average unit dispatching cost of the proposed method in comparison to those of day-ahead dispatching, SO, and RO. The average unit dispatching cost was calculated as the sum of start-up/shutdown costs and output costs, divided by the total number of units.

The comparison results between the proposed method and day-ahead dispatching, SO, and RO are shown in Table 4. It can be observed that the proposed QNN-based complementary scheduling method demonstrates superior economic efficiency compared to the other three methods.

**Table 4.** Comparison of model scheduling costs.

| Title 1 | Multi-Energy Coordination Model Based on QNN | Day-Ahead Dispatching | SO | RO |
|---|---|---|---|---|
| Average cost per unit (JPY/MWh) | 148.7216 | 185.5362 | 155.759 | 201.564 |

## 4. Conclusions

To achieve efficient intraday complementary scheduling of water, wind, and solar energy in a system incorporating multiple clean energy sources, this study proposes a rapid predictive control optimization method based on QNNs for a multi-energy system. The primary objective of this method is to optimize the consumption of renewable energy while coordinating generating units and available flexible loads, facilitating rapid predictive control of intraday dispatching. The key contributions of the proposed method are as follows:

(1) Construction of a multi-energy scheduling model incorporating water, wind, and solar energy, while considering static security constraints and demand-side response.

(2) Pioneering application of QNNs in power grid scheduling, utilizing real-time measurement data from renewable energy and loads to rapidly predict unit outputs and load shedding strategies, serving as an auxiliary model to expedite optimization solutions.

(3) Development of an integrated optimization approach combining QNNs and an optimization model, featuring a closed-loop warm start framework of "initial solution prediction-rolling optimization-feedback evolution" to further enhance the evolution of QNNs and improve predictive control performance.

To validate the efficacy of the proposed model, extensive testing was conducted in the ZD area of the Sichuan power grid. The results demonstrate that the model exhibits commendable performance in terms of solving speed, decision-making accuracy, and economy. Consequently, it can promptly provide reliable auxiliary decision-making control information for intraday dispatching, thereby enhancing the overall efficiency of the system.

**Author Contributions:** Conceptualization, X.Y. and W.W.; methodology, X.Y.; software, Z.C.; validation, X.Y., Z.C. and T.Z.; formal analysis, W.W.; investigation, H.P.; resources, X.Y.; data curation, Z.C.; writing—original draft preparation, X.Y.; writing—review and editing, Z.C.; visualization, T.Z.; supervision, W.W.; project administration, Z.C.; funding acquisition, X.Y. All authors have read and agreed to the published version of the manuscript.

**Funding:** Research Projects of Sichuan New Electric Power System Research Institute, grant number B7199723R005.

**Data Availability Statement:** Data are available on request due to restrictions on privacy or ethics. The data presented in this study are available on request from the corresponding author.

**Acknowledgments:** Thanks to the authors and teachers for their irreplaceable help in this work.

**Conflicts of Interest:** Xi Ye and Tong Zhu were employed by State Grid Sichuan Electric Power Company, Chengdu 610000, Sichuan, China. Zhen Chen and Wei Wei were employed by Electric Power Research Institute, Chengdu 610000, Sichuan, China. The remaining authors declare that the research was conducted in the absense of any commercial or financial relationships that could be construed as a potential conflict of interest.

**Appendix A**

$$V_h^{min} \leq V_{h,t} \leq V_h^{max} \tag{A1}$$

$$P_h^{H\min} \leq P_{h,t}^H \leq P_{h,t}^{H0} \tag{A2}$$

$$Q_h^{Hmin} \leq Q_{h,t}^H \leq Q_h^{Hmax} \tag{A3}$$

$$V_{h,t+1} = V_{h,t} + \left( I_{h,t} - Q_{h,t}^G - Q_{h,t}^C \right) \Delta t \tag{A4}$$

$$I_{h+1,t+\tau} = Q_{h,t}^H + L_{h,t} \tag{A5}$$

$$\delta^L \Delta t \leq \left( P_{h,t+1}^H - P_{h,t}^H \right) \leq \delta^U \Delta t \tag{A6}$$

$$0 \leq P_{r,t}^{IIG} \leq P_{r,t}^{IIG0} \tag{A7}$$

where $V_{h,t}$, $V_h^{\min}$, $V_h^{\max}$ represent the current water volume, minimum storage capacity, and maximum storage capacity of the reservoir, respectively. $P_{h,t}^H$, $P_h^{H\min}$, and $P_{h,t}^{H0}$ represent the current output, minimum active output, and actual maximum output level of cascade hydropower, respectively. $Q_{h,t}^H$, $Q_h^{Hmin}$, and $Q_h^{Hmax}$ represent the current discharge, minimum discharge, and maximum discharge of cascade hydropower, respectively. $I_{h,t}$, $Q_{h,t}^G$, and $Q_{h,t}^C$ are the inflow flow, power generation flow, and discharge flow of the h th reservoir within time t, respectively; $\tau$ and $L_{h,t}$ are the delay factors of interstage flow and interstage inflow of the cascade hydropower system; $\delta^L$ and $\delta^U$ represent the minimum and maximum power change speed of the cascade hydropower system, respectively. $P_{r,t}^{IIG0}$ is the measured data of renewable energy active output.

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
