# Peer review of "Fast Coordinated Predictive Control for Renewable Energy Integrated Cascade Hydropower System Based on Quantum Neural Network"

_electronics, doi:10.3390/electronics13040732_

Round 1

Reviewer 1 Report

Comments and Suggestions for Authors

In my opinion, the idea behind this paper has potential. However, the paper must be considerably improved in order to be considered for publication. In the Introduction/Literature Review section more relevant publications should be cited. Furthermore, the authors should provide graphical representation of all optimization methods that have been used. The authors should include separate paragraph where paper's contributions have been clearly defined. In the current version of the manuscript novelty is questionable. Additionally, in order to improve the dispatch algorithm my suggestion to the authors is to include demand response constraints. Results and Conclusion section should be re-written from scratch in more concise manner.

My suggestion to the editor is a Major Revision.

Comments on the Quality of English Language

English very difficult to understand/incomprehensible

Reviewer 2 Report

Comments and Suggestions for Authors

The manuscript subject “Fast Coordinated Predictive Control for Renewable Energy Integrated Cascade Hydropower System Based on Quantum Neural Network” is interesting. However, the article needs to be improved. In details:

1.      The literature review needs to be improved by adding the information about different models and those that is considering in results such as importance of “Model Solving Time”, “real-time models”, etc.

2.      Section 2, section 3 and lines 323 to 333 can be put in one section as methodology.

3.      Figure 1 need to be updates “Figure 1. Multi hidden layer quantum neural network structure”. inputs, outputs and parameters are not determined, and just layer is appeared in the figure.

4.      The “comparison between the QNN model and the BP” in 4.1 is not mentioned as the study objective, and just appeared as a result. The lines 94 to 105 is better to be rewritten, or at a new section as methodology the research flowchart be added.

5.      “Figure 6. Model optimization scheduling results” needs a color legend or description. The same for Figure 7.

6.      The name of section 4.5 seems incorrect. It is cost analysis not model cost. More details about dispatching cost calculations need to be added.

7.      Line 421: “average unit dispatching cost” a reference for this value that is not calculated in this study and just used for comparisons seems necessary.

Suggestions:

8.      Most of the equations are general and no need to be inside the main text and can be put in an appendix.

Round 2

Reviewer 2 Report

Comments and Suggestions for Authors

Suggestion:

Section 2 can be combined with lines 313 to 323 as methodology.

Author Response

We appreciate your guidance on this paper once again and value your valuable feedback. We have carefully reviewed the references and updated some of them. The revised references have been highlighted in red in the manuscript's reference section. Additionally, we have supplemented content related to demand response(DR), enriching the manuscript. We considered load shedding as the type of flexbile load, and added relevant optimization objectives, constraints. New case visualizations have also been added in the validation, specifically in Figure 7(c) in the manuscript. Moreover, we have combined section 2 with lines 313 to 323 as methdology.